# The Effects of Rock Index Tests on Prediction of Tensile Strength of Granitic Samples: A Neuro-Fuzzy Intelligent System

**Yan Li** [1], **Fathin Nur Syakirah Hishamuddin** [2], **Ahmed Salih Mohammed** [3], **Danial Jahed Armaghani** [4,*], **Dmitrii Vladimirovich Ulrikh** [4], **Ali Dehghanbanadaki** [5] **and Aydin Azizi** [6]

1    Art College, Chongqing Technology and Business University, Xuefu Avenue, Nan'an District, Chongqing 4000678, China; 13272855916@163.com
2    Department of Civil Engineering, Faculty of Engineering, University of Malaya, Kuala Lumpur 50603, Malaysia; fathinnursyakirah@siswa.um.edu.my
3    Civil Engineering Department, College of Engineering, University of Sulaimani, Sulaymaniyah 46001, Iraq; ahmed.mohammed@univsul.edu.iq
4    Department of Urban Planning, Engineering Networks and Systems, Institute of Architecture and Construction, South Ural State University, 76, Lenin Prospect, 454080 Chelyabinsk, Russia; ulrikhdv@susu.ru
5    Department of Civil Engineering, Damavand Branch, Islamic Azad University, Damavand 39718-78911, Iran; A.dehghanbanadaki@damavandiau.ac.ir
6    School of Engineering, Computing and Mathematics, Wheatley Campus, Oxford Brookes University, Oxford OX33 1HX, UK; aydin.azizi@brookes.ac.uk
*    Correspondence: danialarmaghani@susu.ru

**Abstract:** Rock tensile strength (TS) is an essential parameter for designing structures in rock-based projects such as tunnels, dams, and foundations. During the preliminary phase of geotechnical projects, rock TS can be determined through laboratory works, i.e., Brazilian tensile strength (BTS) test. However, this approach is often restricted by laborious and costly procedures. Hence, this study attempts to estimate the BTS values of rock by employing three non-destructive rock index tests. BTS predictive models were developed using 127 granitic rock samples. Since the simple regression analysis did not yield a meaningful result, the development of models that integrate multiple input parameters were considered to improve the prediction accuracy. The effects of non-destructive rock index tests were examined through the use of multiple linear regression (MLR) and adaptive neuro-fuzzy inference system (ANFIS) approaches. Different strategies and scenarios were implemented during modelling of MLR and ANFIS approaches, where the focus was to consider the most important parameters of these techniques. As a result, and according to background and behaviour of the ANFIS (or neuro-fuzzy) model, the predicted values obtained by this intelligent methodology are closer to the actual BTS compared to MLR which works based on linear statistical rules. For instance, in terms of system error and a-20 index, values of (0.84 and 1.20) and (0.96 and 0.80) were obtained for evaluation parts of ANFIS and MLR techniques, which revealed that the ANFIS model outperforms the MLR in forecasting BTS values. In addition, the same results were obtained through ranking systems by the authors. The neuro-fuzzy developed in this study is a strong technique in terms of prediction capacity and it can be used in the other rock-based projects for solving relevant problems.

**Keywords:** rock strength; tensile behaviour; neuro-fuzzy; regression; non-destructive tests

## 1. Introduction

In rock engineering, rock fracture mechanics are associated with the behaviours of rock deformation and failure patterns caused by crack initiation and propagation. The growth of cracks in rocks happens due to small micro-cracks, micro-defects, and failures of large pre-existing fractures in rock [1]. Rock has lower tensile resistance compared to compressive and shear resistance due to its brittleness properties. Therefore, understanding rock

behaviour such as tensile properties is essential for solving geotechnical problems during underground openings, surface excavation, and rock blasting and to ensure underground cavern stability. There are many methods for predicting rock tensile strength (TS). Direct test is considered the most effective method to derive the tensile capacity of rock specimen. Direct TS value can be determined accurately using a dumbbell-shaped specimen [2]. However, difficulties are often associated with the direct tensile test. Indirect approaches such as the Brazilian disc test are widely utilised by researchers due to their simplicity and efficiency during sample preparation and testing procedures [3]. Valid tensile pattern Brazilian disc of rock specimens can also be visualised through digital image correlation [4]. Several tests such as the half-ring and semi-circular bending tests are relevant to determine the TS of brittle rocks [5,6].

Aiming to obtain the most reliable method of TS value prediction, Xia et al. [7] examined the dynamic TS of Laurentian granite three ways, by the dynamic direct tension test (DTT), dynamic Brazilian test (BT), and dynamic semi-circular bend test (SCB). Their findings suggest that the overestimation of TS value for BT and DTT can be corrected using overload and internal friction impact mechanisms. The flat-joint model can reflect the peak tensile stress in Brazilian disc specimens [8]. Yuan and Shen [9] proposed improving the Brazilian tensile strength (BTS) method by increasing the number of disk specimens to twice the number of standard samples. Larger specimen size will underestimate the intrinsic rock TS as an indentation type of failure mechanism is anticipated [10]. The behaviour of rock in tension can be an effective indicator of the state of rock weathering. Aydin and Basu [11] stated that the Brazilian deformation index could differentiate rock weathering grade and demonstrate the distinct behavioural pattern of rock during the weathering process. Additionally, rock size contributes to underestimating TS value, while rock heterogeneity leads to an overestimation of TS [12]. Thus, extensive correction coefficient studies are needed to estimate rock tensile strength, ideally by using the Brazilian testing method [13].

Several experiments have been performed to estimate the rock TS reliably. To mention a few examples, Nazir et al. [14] and Kabilan [15] have presented the significant correlation between BTS and unconfined compressive strength in rocks statistically. Simple regression modelling revealed that input parameters such as point load index ($Is_{50}$), dry density ($DD$), and Schmidt hammer rebound number ($R_n$) gave an average level of accuracy to estimate the BTS values. Table 1 presents some of the important proposed empirical equations to estimate BTS values together with their regression types and performance predictions. Although performance capacities of these techniques are quite suitable, they are often unpredictable when some uncertainties are not addressed during the development process. However, numerous studies have highlighted the outstanding results of some new computational techniques, i.e., artificial intelligence in evaluating and predicting rock strength values [16,17].

**Table 1.** Some of the proposed empirically correlations between *BTS* and other rock tests.

| References | Proposed Equations | Regression Type | $R^2$ | Description |
|---|---|---|---|---|
| Heidari et al. [18] | $BTS = 0.88Is_{50} + 2.70$ | L | 0.9 | 40 Gypsum rocks |
| Altindag and Guney [19] | $BTS = 0.0423SH^{1.2799}$ | NL | 0.8 | 143 rock samples |
| Farah [20] | $UCS = 12.308BTS^{1.0725}$ | NL | 0.6 | 195 of limestone specimens |
| Kahraman et al. [21] | $UCS = 10.61BTS$ | L | 0.5 | Igneous rocks |
| Nazir et al. [14] | $UCS = 9.25BTS^{0.947}$ | NL | 0.9 | 40 laboratory strength tests on dry limestone |
| Mohamad et al. [22] | $UCS = 15.361BTS - 10.303$ | L | 0.8 | 40 sets soft rock samples |
| Khandelwal et al. [23] | $BI = 0.59BTS^{0.769} - 5.085BTS^{0.531} + 0.009\gamma^{2.332}$ | NL | 0.9 | 13 types of rock from USA |
| Mahdiyar et al. [24] | $BTS = 1.993e^{0.027R_n}$ | NL | 0.7 | 100 granite block samples |
| Mahdiyar et al. [24] | $BTS = 0.022e^{2.182DD}$ | NL | 0.7 | 100 granite block samples |

$BTS$ = Brazilian tensile strength, $Is_{50}$ = point load index, $R_n$ = rebound number, $\gamma$ = unit weight, DD = dry density, $UCS$ = uniaxial compressive strength, $BI$ = brittleness index, $SH$ = surface hardness, L = linear, NL = non-linear, $R^2$ = coefficient of determination.



Artificial intelligence approaches perform the automatic creation of an analytical model that recognizes patterns and make decisions without human interventions. Many researchers have emphasised the capabilities of these techniques in the field of geotechnical engineering, and they have proven to aid various civil and mining engineering problems [25–40]. For forecasting BTS values, Singh et al. [41] performed artificial neural network (ANN) modelling analysis on schistose rock samples and they reported a good level of prediction performance. Çanakci et al. [42] evaluated the performance of the ANN and Gene Expression Programming model in predicting rock tensile and compressive strength. The mechanical properties of Yavuzeli basaltic rocks from a region in Turkey were used to construct the model algorithm. The neural network algorithm obtained better results in terms of coefficient of determination ($R^2$) with a value of 0.829. Ceryan et al. [43] did a thorough analysis of rock TS modelling using support vector machine (SVM) approaches, the least square SVM method, and ANN to weigh their computational advantage. Finally, they introduced LS-SVM as a robust model that can accurately and efficiently predict rock TS because the analysis process is much faster than the other two models. Table 2 shows some of the important studies in the areas of BTS prediction using different artificial intelligence approaches. As is obvious from this table, the artificial intelligence techniques are able to provide higher capability levels compared to empirical techniques in estimating BTS values.

**Table 2.** Artificial intelligence approaches presented to estimate rock BTS values.

| References | Model | Input Parameters | Model Performance | Description |
|---|---|---|---|---|
| Singh et al. [41] | ANN | Petrographical characteristics | MAPE = 11% | Schistose rocks |
| Çanakci et al. [42] | ANN | $V_p$, DD, $R_n$, WA | $R^2$ = 0.99 | 86 samples of basalt from Turkey |
| Gurocak et al. [44] | MLPN | $Is_{50}$, $R_n$, $\gamma$ | $R^2$ = 0.84 | 174 samples from Turkey |
| Ceryan et al. [43] | LS-SVM | POR, $V_p$, SDI, aggregate impact | $R^2$ = 0.86 | 55 carbonate rocks from Turkey |
| Mahdiyar et al. [24] | PSO-ANN | $Is_{50}$, DD, $R_n$ | $R^2$ = 0.93 | Granite rock samples |
| Huang et al. [45] | IWO-ANN | $Is_{50}$, DD, $R_n$ | $R^2$ = 0.92 | 100 granite samples |

$V_p$ = p-wave velocity, $Is_{50}$ = point load index, DD = dry density, $R_n$ = rebound number, WA = water absorption, $\gamma$ = unit weight, MAPE = mean absolute percentage error, MLPN = multilayer perceptron network, PSO = particle swarm optimization, IWO = invasive weed optimisation.

The problem related to the difficulty of conducting BTS tests, as mentioned before, can be solved using rock index tests, which are easier and faster to carry out. The focus of previous studies was to investigate the effects of both destructive and non-destructive rock index tests in estimating BTS values. However, sometimes there is a need to have non-destructive tests which will not fail during or after the test. Therefore, the objective of this study is to consider and use results of only non-destructive tests, i.e., ultrasonic velocity, Schmidt hammer, and density for prediction of BTS values. To do this, different basic and advanced statistical models, together with an adaptive neuro-fuzzy inference system (ANFIS) intelligent technique, are proposed for tensile strength prediction. The models, their backgrounds, design procedures, and the obtained results in evaluating behaviour of rock tensile strength will be discussed in detail. The more accurate and reliable model will be introduced for the same purpose.

## 2. Methods and Material

### 2.1. Laboratory Tests

One hundred fifty-four rock samples of blocks of granite were brought from a tunnel project located in Malaysia for assessment. In this project, there were three tunnel boring machines, namely Kamila, Selpah, and Tiara Midori. The tunnel is intended to help mitigate potential water shortages in the problematic region. The project utilized the available surface water runoffs from several important rivers, i.e., Kelau River, Bentong

River, and Telemong River. The tunnel starts in Pahang state with length of about 45 km and goes to Selangor state, with different overburden values in its route.

Rock index tests can be categorized into destructive and non-destructive tests. Destructive tests like point load are conducted to the specimen's failure to understand the sample behaviour under failure loading and stage, while non-destructive tests such as the Schmidt hammer are those without damage during and after tests conducted to evaluate a particular group behaviour, such as physical characteristics of the samples. This study focuses on the use of only non-destructive tests, i.e., ultrasonic velocity, Schmidt's hammer rebound, and density in assessment and evaluation of tensile response of the rock samples. Hardness of the rock surface was measured by performing non-destructive testing known as the Schmidt Hammer Rebound (SHR) test. Following the testing procedure found in the International Society for Rock Mechanics (ISRM) [46] guidelines, the average reading computed from 10 SHR tests was denoted as $R_n$ rebound number. The L-type hammer was mounted vertically downwards against the rock samples. In addition, an ultrasonic velocity test was conducted to measure the degree of compactness of rock material. The core samples should be flat at both ends to transmit primary waves (p-wave) through the core samples. This test was conducted four times using Portable Ultrasonic Non-Destructive Digital Indicating Tester (PUNDIT) equipment following the ISRM [46] guidelines. The values recorded from the PUNDIT equipment are denoted as $V_p$. Specimens with higher density (lesser voids) display a higher $V_p$ value. Apart from $R_n$ and $V_p$, dry density (DD) tests were performed to measure the rock's physical properties.

Brazilian tests were performed to measure the TS of rock samples indirectly. Cylindrical disc-shaped specimens with flat end surfaces were prepared prior to the testing process. According to ISRM [46], the specimen's size should have an approximate thickness/diameter ratio of 2. In this article, a total of 127 data samples were established for the modelling and analyses, where DD, $V_p$, and $R_n$ were set as predictors and BTS was assigned as the target value, which is very important to accurately predict.

The laboratory test results for this this study are summarized in Table 3.

**Table 3.** Laboratory test results summary.

| Parameters Symbol | Group | Unit | Min | Max | Ave. | Sd. |
|:---:|:---:|:---:|:---:|:---:|:---:|:---:|
| $R_n$ | Input | - | 20 | 61 | 40.5 | 9.93 |
| $V_p$ | Input | m/s | 2643 | 7702 | 5172.5 | 1331.60 |
| DD | Input | g/cm$^3$ | 2.35 | 2.79 | 2.57 | 0.11 |
| BTS | Output | MPa | 3.2 | 12.9 | 8.05 | 2.58 |

Min = minimum value, Max = maximum value, Ave = average value, and Sd. = standard deviation.

### 2.2. ANFIS Background

ANFIS is a proper intelligent system that integrates fuzzy logic with the principle of neural networks [47]. Such a framework makes ANFIS modelling more systematic and less reliant on expert knowledge [48]. Hence, this section will describe ANFIS architecture and learning algorithms for the Takagi-Sugeno-Kang (TSK) fuzzy model. Generally, ANFIS efficiency is influenced by the selection of number and shape of membership function (MF), the number of rules, and their learning techniques. Seven fuzzy MFs are integrated in the ANFIS tools while four of them are the most widely applied. The four types of MF are Gaussian combination (gauss2mf), Gaussian curve (gaussmf), bell shape (gbellmf), and trapezoidal shape (trapmf). The output MFs for the TSK-model consist of two parameters which are constant and linear.

Five layers (i.e., fuzzy layer, product layer, normalised layer, defuzzy layer, and total output layer) are needed to construct this interference system. The ANFIS algorithm system can be simplified by assuming two inputs ($x$ and $y$) and one output, $f$. The if-then rules for the first order of TSK fuzzy model may be conveyed as [48–50]:

$$Rule\ 1: If\ (x\ is\ A_1)\ and\ (y\ is\ B_1),\ then\ z\ is\ f_1(x,y) \tag{1}$$

$$Rule\ 1 : If\ (x\ is\ A_2)\ and\ (y\ is\ B_2),\ then\ z\ is\ f_2(x,y) \tag{2}$$

where $x$ and $y$ are the input of ANFIS, $A$ and $B$ are the nonlinear parameters, and $f_i(x,y)$ is the output of ANFIS expressed in terms of first-order polynomial. A five-layer ANFIS with three inputs and one output model structure will be used to estimate BTS. The nodes function for each layer will be further elaborated as follows:

- Layer 1 (fuzzy layer): Comprises adaptive nodes with functions expressed (Equations (3) and (4)) as:

$$O_{1,i} = \mu_{Ai}(x),\ i = 1,2 \tag{3}$$
$$O_{1,j} = \mu_{Bj}(y),\ j = 1,2 \tag{4}$$

where $x$ and $y$ indicate the input nodes, $A_i$ and $B_i$ denotes the linguistic labels, $\mu(x,y)$ implies the MFs.

- Layer 2 (product layer): Includes the product layer of two fixed nodes labelled $\Pi$ expressed as Equation (5).

$$O_{2,i} = \omega_i = \mu_{Ai}(x) \times \mu_{Bi}(y)\ ,\ i = 1,2 \tag{5}$$

- Layer 3 (normalised layer): Node function is to normalise the weight function of the following process and is labelled as $N$, Equation (6):

$$O_{3,1} = \overline{\omega}_i = \frac{\omega_i}{\omega_1 + \omega_1},\ (i = 1,2) \tag{6}$$

- Layer 4 (defuzzy layer): Contains adaptive nodes marked by a square, Equation (7):

$$O_{4,i} = \overline{\omega}_i f_i,\ (i = 1,2) \tag{7}$$

- Layer 5 (total output layer): Contains fixed node with function to compute overall output, Equation (8):

$$O_{5,i} = f_{out} = \sum \overline{\omega}_i f_i \tag{8}$$

where $O_{1-5,i}$ denotes the output of each layer and $\omega_i$ represents the weight function of the next layer. Figure 1 illustrates the architecture of ANFIS. In addition, the overview of ANFIS flowchart is illustrated in Figure 2.

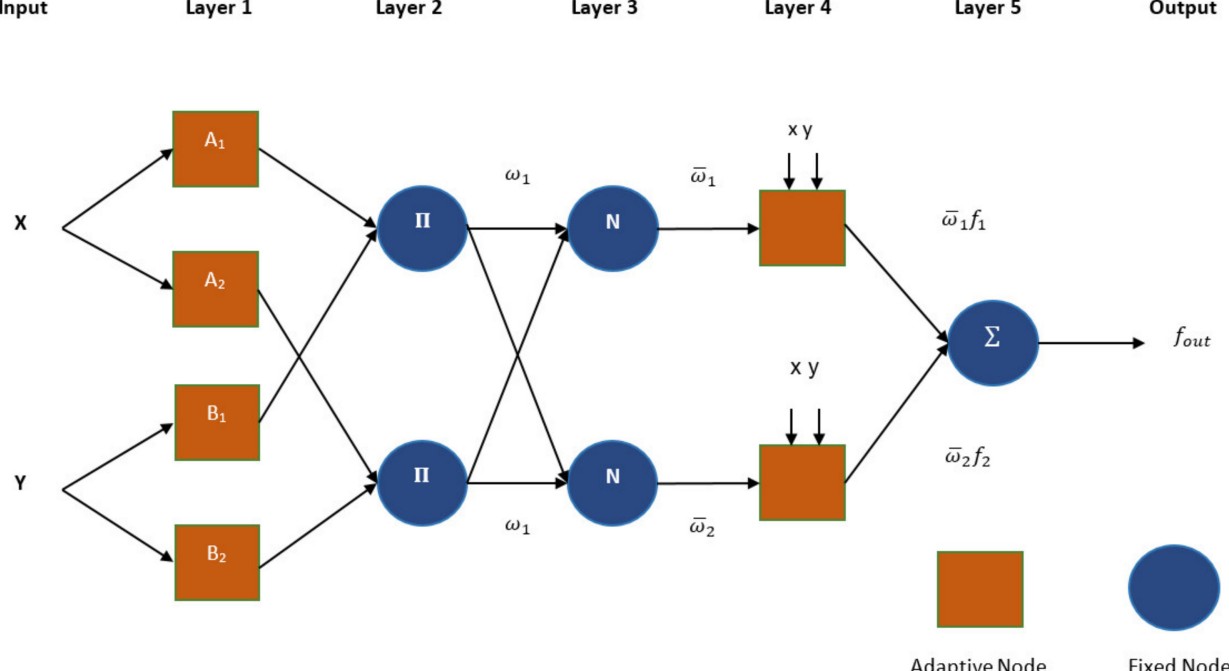

**Figure 1.** ANFIS architecture.

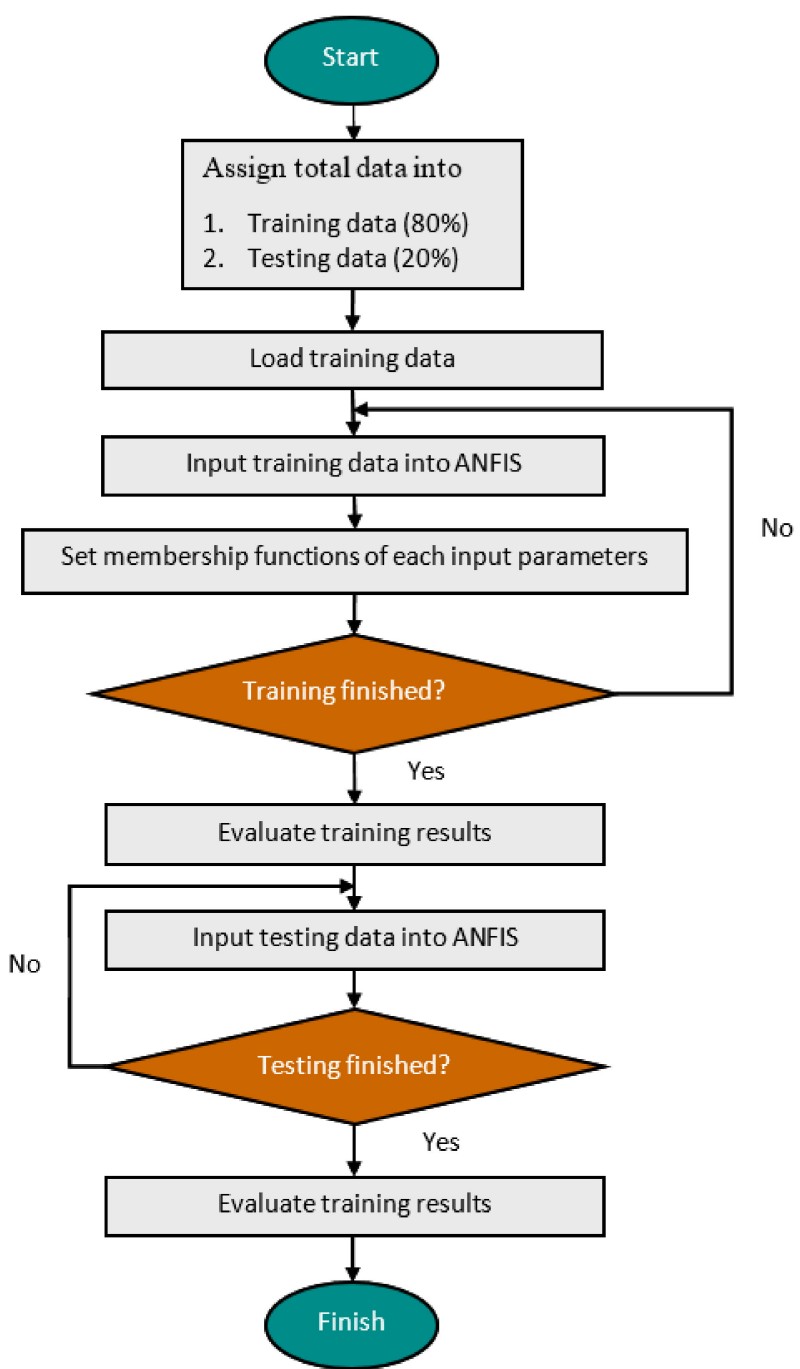

**Figure 2.** ANFIS flowchart for prediction purposes.

### 2.3. Step-by-Step Overview of Research

The general flowchart used for developing the rock TS model is shown in Figure 3. First, literature reviews of relevant research papers regarding indirect TS prediction were conducted. After identifying the research objective and problem statement, the Pahang-Selangor tunnel project was chosen to be studied. Obtaining adequate empirical data from related literature helped in the selection of the most influential parameters. In order to build the model, the database of non-destructive tests was prepared. In each model, the considered input parameters were $R_n$, $V_p$, and DD, while BTS was set as the target parameter. Then, both mathematical and soft-computing methods are used in this study to evaluate tensile behaviour of rock material. The empirical equations for TS estimation are proposed using simple regression (SR) and multiple linear regression (MLR).

Following that, ANFIS modelling is conducted to predict the rock TS values. Each model utilized the mentioned parameters and evaluated them based on their predictor intervals performance. For comparison purposes, performance indices are applied to the proposed model, respectively. Finally, the most reliable model is introduced as the suitable indirect approach for predicting TS rock.

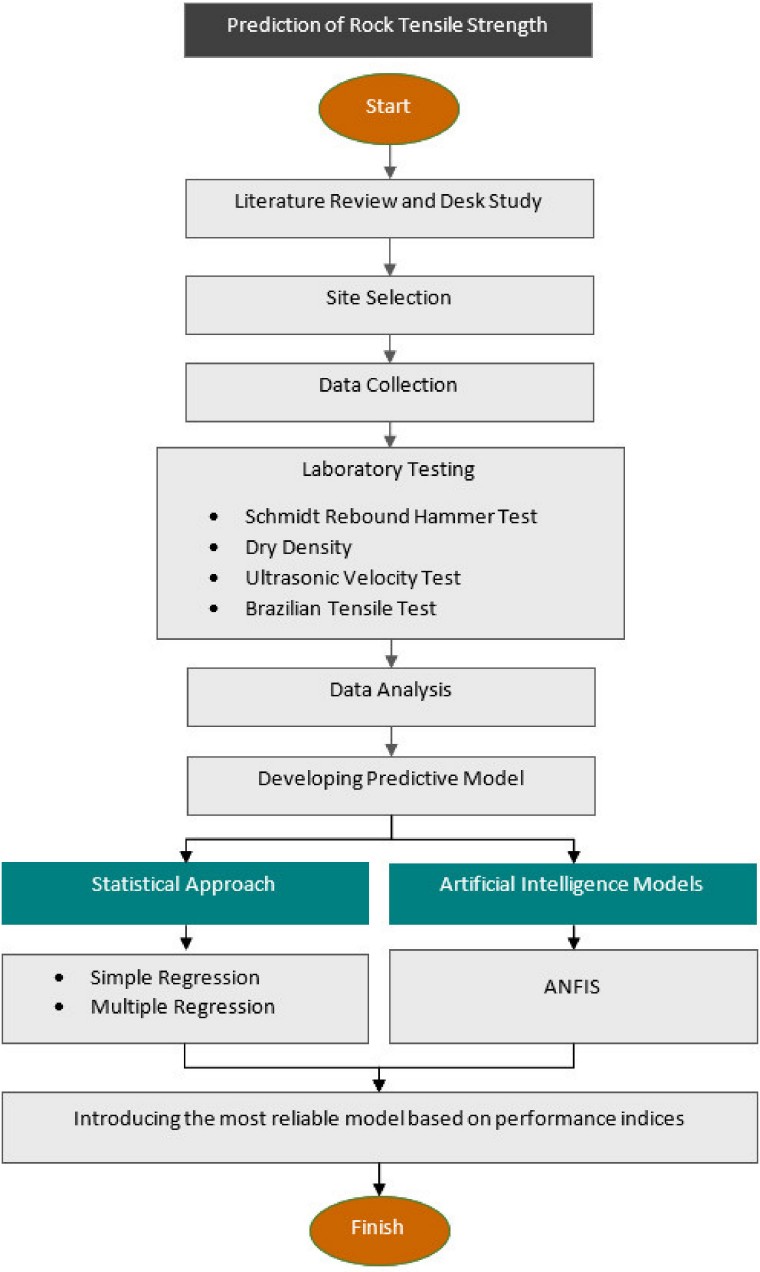

**Figure 3.** The sequence of research methodology.

It is important to note that we followed a typical flow of simulation and prediction studies in the area of rock mechanics [16,51]. They normally start with a SR technique, which is easy to conduct, but it is at the same time not accurate enough to solve the problems. Then, we move to a MLR model where more than one predictor can be used to increase prediction capacity. Finally, to increase the performance capacity, an intelligence technique (which is AFINS in this study) is used to predict the target values (which is BTS in this study).

### 2.4. Statistical Index

In this investigation, $R^2$, root mean square error (*RMSE*), variance account for (VAF%), and a-20 index were selected and used to assess the predictive models. Willmott and Matsuura [52] stated that large errors influenced the total square error rather than the smaller error. When the variances associated with the frequency distribution of error magnitudes increases, the *RMSE* increases. The fit accuracy of *RMSE* is numerically equivalent to zero. The *RMSE* formula is based on Equation (9).

$$RMSE = \sqrt{\frac{1}{n} \sum_{i=1}^{N} (y' - y)^2} \tag{9}$$

where $y$, $y'$ and $N$ conveys the measured, predicted, mean values, and the total numbers of data, respectively. $R^2$ is an indicator to determine the model fit for a set of quantitative dependent variables and their relation to the dependent variable. The determination of acceptable $R^2$ value is important to assess the adequacy and efficiency of the regression model. According to Menard [53], a "good" $R^2$ statistic should be dimensionless, have well defined values with endpoints that lead to a perfect fit to lack of fit range ($0 \leq R^2 \leq 1$), and be applicable to any models with random or non-random variables. The $R^2$ measurements in this study are based on Equation (10).

$$R^2 = 1 = \frac{\sum_i (y_i - \hat{y}_i)^2}{\sum_i (y_i - \bar{y})^2} \tag{10}$$

where $y_i$ indicates the measured value of the dependent variable with a value between zero to one. $\hat{y}_i$ and $\bar{y}$ conveys the predicted and mean values of dependent variables, respectively. *VAF* is known as the proportion of the total population of the dependent variable that can be clarified by the factor of interest. Equation (11) can be used to describe the *VAF* formula:

$$VAF = \left[ 1 - \frac{var(y - y')}{var(y)} \right] \times 100\% \tag{11}$$

where observed, predicted, and mean values are represented by $y$ and $y'$, respectively. According to Xu et al. [54], a20-index is the newly proposed engineering index that is beneficial for evaluating artificial intelligence models by showing the number of samples that fit the prediction values with a deviation of ±20% compared to experimental values, as presented in Equation (12).

$$a20 - index = \frac{m^{20}}{M} \tag{12}$$

where $M$ represents the amount of dataset samples and $m^{20}$ denotes the rate of experimental value/predicted value that lies between the range of 0.80 to 1.20.

### 3. Modelling, Analysis, and Results

### 3.1. SR Modelling

The SR technique can evaluate the correlation between two variables (predictors and targets parameters). SR assumes that the data followed a normal distribution pattern. In this study, SR analysis was performed for 127 data samples during the initial stage of the rock TS prediction. This part of the study was conducted to identify the relationship between each dependant variable (as mentioned in Table 3) and the BTS values as a target of the study. Various types of equations such as exponential ($y = mex^{cx}$), linear ($y = mx + c$), logarithmic ($y = m \ln(x) + c$), and power ($y = mx^c$) were used and the data samples were evaluated for each model to obtain the best empirical forecasting outcomes. Statistically, the best-fitted regression model results should project the value of $R^2 = 1$. Table 4 presents the results of each SR model analysis (according to the type of equations) and their ranking. Noted that for each SR model, rank 4 indicates the best correlation among all equations proposed.

**Table 4.** Results of SR model analysis in estimating rock strength values.

| Model | Input | Equation Type | Equation | $R^2$ | Rank |
|-------|-------|---------------|----------|-------|------|
| 1 | $R_n$ | Exponential | $BTS = 2.5284e^{0.0279R_n}$ | 0.657 | 1 |
|   |       | Linear | $BTS = 0.2182R_n - 0.5659$ | 0.704 | 4 |
|   |       | Logarithmic | $BTS = 8.4797ln(R_n) - 22.854$ | 0.690 | 3 |
|   |       | Power | $BTS = 0.139R_n^{1.0984}$ | 0.659 | 2 |
| 2 | $V_p$ | Exponential | $BTS = 2.4945e^{0.0002V_p}$ | 0.646 | 4 |
|   |       | Linear | $BTS = 0.0015V_p - 0.3164$ | 0.638 | 2 |
|   |       | Logarithmic | $BTS = 7.518ln(V_p) - 56.298$ | 0.611 | 1 |
|   |       | Power | $BTS = 0.0013V_p^{1.0176}$ | 0.638 | 2 |
| 3 | DD | Exponential | $BTS = 0.0101e^{2.5761DD}$ | 0.689 | 3 |
|   |    | Linear | $BTS = 19.171DD - 41.274$ | 0.670 | 1 |
|   |    | Logarithmic | $BTS = 49.169ln(DD) - 38.371$ | 0.671 | 2 |
|   |    | Power | $BTS = 0.0146DD^{6.6245}$ | 0.693 | 4 |

From Table 4, Model 1 shows a lack of fit approximation with a similar $R^2$ value of $0.658 \pm 0.001$ for exponential and power functions, respectively. In contrast, for Model 2, exponential function indicates a good correlation between $V_p$ and BTS, with an $R^2$ of 0.646. It can be seen that both linear and power functions have the same ranking score (2) while logarithmic function ranks the lowest (1). Meanwhile, for Model 3, the power function has the second highest correlation coefficient ($R^2 = 0.693$) after Model 1. Therefore, the highest performance prediction equations were selected for each predictor (i.e., linear for $R_n$, exponential for $V_p$ and power for DD). The graph of data plot for the best approximation equations for $R_n$, $V_p$ and DD are shown in Figure 4. Overall, the scatter plot of all models has positive slopes, which corresponds to the positive correlation values. SR analyses indicate that Model 1 with $R_n$ input has the strongest correlation value ($R^2 = 0.70$) among two other models. By referring to previous investigations [24], the results of $R^2$ for SR analyses fall within the range of 0.5 to 0.81. In this study, the average $R^2$ for SR analysis was 0.664, which is satisfactory. However, consideration of only one parameter for the prediction model is not enough to get the highest degree of accuracy. Hence, in the next prediction stage, the MLR modelling will be applied where more than one input is incorporated to predict BTS values.

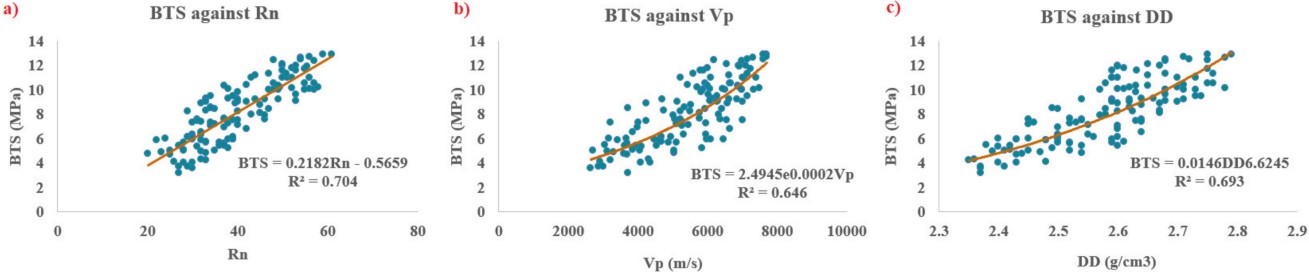

**Figure 4.** SR analysis results (**a**) BTS vs. $R_n$, (**b**) BTS vs. $V_p$, and (**c**) BTS vs. DD.

### 3.2. MLR Modelling

Generally, regression analysis requires a strong correlation between the variable to validate its credibility. In essence, MLR is the extension of ordinary least-squares regression that requires more than one independent variable. In the next stage of regression analysis, four models are introduced to predict *BTS*. Each model integrates different numbers of inputs but use the same output (i.e., *BTS*). Previous studies have mentioned that higher degree of accuracy can be achieved when more than one input is being considered [55,56]. A straightforward ranking approach can be used to evaluate each model [57]. The most fitting model for MLR analysis is the one that generates the highest-ranking score. By

referring to previous studies [58,59], 80% (102) of the whole data samples (127) were selected randomly for the model's training purposes. The remaining 20% (25) were used to test the models. Table 5 shows the MLR regression equations for different models with various combination types.

**Table 5.** MLR equations for BTS prediction.

| Model No. | Input | Equation |
|-----------|-------|----------|
| MR1 | $V_p$, DD | $BTS = 0.00089V_p + 11.5223DD - 26.4453$ |
| MR2 | $R_n$, DD | $BTS = 0.13804R_n + 10.5113DD - 24.5321$ |
| MR3 | $R_n$, $V_p$ | $BTS = 0.143231R_n + 0.00076V_p - 1.76855$ |
| MR4 | $R_n$, $V_p$, DD | $BTS = 0.11068R_n + 0.00041V_p + 8.69973DD - 20.9934$ |

The series of MLR analyses for training and testing datasets together with prediction capacity indices results are presented in Tables 6 and 7, respectively. The testing $R^2$ values for MLR Model 1 (MLR1) which integrates two inputs appears to lie in satisfactory values of 0.68. Meanwhile, models MLR2 to MLR4 exhibit almost similar determination coefficient values ($R^2$ = 0.79, 0.71, and 0.78) that are acceptable. Considering the results of both train and test stages, it can be concluded that the MLR4 model reveals the highest total ranking score (15 + 13 = 28) for indices comparison with *RMSE*, *VAF*, $R^2$, and a20-index values of 1.051, 83.564%, 0.836, 0.843 for training datasets and 1.201, 77.869%, 0.780 and 0.80 for testing datasets. Hence, the recommended regression equation for MLR analysis is in fact the last equation presented in Table 5. Later, the performance of MLR models will be discussed in more detail.

**Table 6.** Training results of MLR analysis.

| Model No. | Training Data | | | | Training Ranking | | | | Total Rank |
|-----------|------|---------|-------|----------|------|-----|-------|----------|------------|
| | RMSE | VAF (%) | $R^2$ | a20-Index | RMSE | VAF | $R^2$ | a20-Index | |
| MR1 | 1.24 | 76.97 | 0.77 | 0.75 | 1 | 1 | 1 | 1 | 4 |
| MR2 | 1.09 | 82.17 | 0.82 | 0.84 | 3 | 3 | 3 | 3 | 12 |
| MR3 | 1.22 | 77.75 | 0.78 | 0.75 | 2 | 2 | 2 | 1 | 7 |
| MR4 | 1.05 | 83.56 | 0.84 | 0.84 | 4 | 4 | 4 | 3 | 15 |

**Table 7.** Testing results of MLR analysis.

| Model No. | Testing Data | | | | Testing Ranking | | | | Total Rank |
|-----------|------|---------|-------|----------|------|-----|-------|----------|------------|
| | RMSE | VAF (%) | $R^2$ | a20-Index | RMSE | VAF | $R^2$ | a20-Index | |
| MR1 | 1.43 | 67.42 | 0.68 | 0.72 | 1 | 1 | 1 | 1 | 4 |
| MR2 | 1.18 | 78.94 | 0.79 | 0.76 | 4 | 4 | 4 | 3 | 15 |
| MR3 | 1.39 | 70.13 | 0.71 | 0.72 | 2 | 2 | 2 | 1 | 7 |
| MR4 | 1.20 | 77.87 | 0.78 | 0.80 | 3 | 3 | 3 | 4 | 13 |

### 3.3. ANFIS Modelling

As mentioned in the last section, the modelling by ANFIS should be started using trained and test data that are already divided. In ANFIS modelling, the most important factors/values should be considered. The MFs in ANFIS modelling can be customised accordingly as one of the most important ANFIS parameters. According to MLR analyses, Model 4, which takes into account three inputs, outperforms the other MLR models. In this regard, the general characteristics of the ANFIS model consisted of three inputs data (including $R_n$, $V_p$, DD) and one output data (i.e., BTS). The size of 102 × 4 training and 25 × 4 testing data were imported in the workspace and loaded into the system. Initially, various fuzzy-interference system (FIS) properties were applied to define the acceptable ANFIS architecture basis. The FIS model structure set to "grid partition" was used to

classify the data and activate the neuro-fuzzy designer dialogue box. By utilising the same sequence of training and testing datasets as the MLR study, the constructed ANFIS models were educated with three types of MFs, namely Gaussian curve (gaussmf), Gaussian combination (gauss2mf), and generalized bell-shape (gbellmf). In addition, the model was constructed separately for each MF number (i.e., 2, 3, 4, and 5) and output type (i.e., constant and linear). Then, the fitness of each model with various ANFIS architectures were weighed against one another according to their statistical index results.

During the ANFIS modelling, a total number of 24 different models were developed to predict BTS values. However, to ensure that the prediction models are optimized and that the statistical data are not overfitted or underfitted, the chosen models should not have a substantial difference in training and testing performance. Hence, by following these benchmarks, the strength of remaining 8 ANFIS models were emphasized in this study and the models' specifications are presented in Table 8. Overfitting is a frequent issue in ANFIS modelling that appears when ANFIS overtrains the data [60]. The training of datasets using ANFIS has a maximum number of epochs before overfitting takes place which results in inaccurate prediction output. Multiple iteration guesses can be used to determine the optimal number of epochs. At first, to decide the minimum number of training epochs for each model, iteration was set up to 100 epochs, except for Model 1, which had its epoch increased up to 150. An optimal range of epochs value was created when the value of error tolerance became constant. The fitness of all ANFIS models was examined through evaluating their *RMSE*, *VAF*, $R^2$, and a20-index. A ranking system [61] was applied to each model to identify the model's performance in predicting BTS. The statistical indices computed and ranking scores for the ANFIS training and testing datasets are shown in Table 9. Based on this table, Model 4 was established as the most robust model in predicting BTS as it indicates the highest total ranking scores of (32 + 32 = 64 for train and test stages). Among all eight developed models, the statistical indices computed for Model 4 show significant improvement for training (*RMSE* = 0.69, *VAF* = 92.99%, $R^2$ = 0.93, a20-index = 0.96) and testing datasets (*RMSE* = 0.74, *VAF* = 91.62%, $R^2$ = 0.92, a20-index = 0.96) results. Figure 5 shows the base FIS and the proposed ANFIS structure for the selected model. Additionally, more information regarding the ANFIS parameters for Model 4 (i.e., the best model) is presented in Table 10. Gaussian MFs of the input parameters including $R_n$, $V_p$, DD are also displayed in Figure 6. These MFs and their range will give a better view to the readers or researchers when they wish to solve similar problems using ANFIS. More discussion regarding the best ANFIS model will be provided later.

**Table 8.** Eight ANFIS models and their specifications in predicting BTS values.

| Model Name | Input | | Output | Epoch |
|---|---|---|---|---|
| | MF No. | MF Type | MF Type | |
| Model 1 | 222 | Bell Membership | Constant | 117 |
| Model 2 | 333 | Bell Membership | Constant | 10 |
| Model 3 | 222 | Gaussian | Constant | 23 |
| Model 4 | 333 | Gaussian | Constant | 88 |
| Model 5 | 222 | Gaussian 2 | Constant | 17 |
| Model 6 | 333 | Gaussian 2 | Constant | 6 |
| Model 7 | 222 | Bell Membership | Linear | 9 |
| Model 8 | 222 | Gaussian 2 | Linear | 10 |

**Table 9.** Statistical indices for testing and training ANFIS models.

| Model Name | Training Datasets | | | | | | | | | Testing Datasets | | | | | | | | |
|---|---|---|---|---|---|---|---|---|---|---|---|---|---|---|---|---|---|---|
| | Statistical Index | | | | Rank | | | | Total | Statistical Index | | | | Rank | | | | Total |
| | RMSE | VAF (%) | $R^2$ | a20-Index | RMSE | VAF | $R^2$ | a20-Index | | RMSE | VAF (%) | $R^2$ | a20-Index | RMSE | VAF | $R^2$ | a20-Index | |
| Model 1 | 0.98 | 85.69 | 0.86 | 0.89 | 3 | 3 | 5 | 5 | 16 | 1.10 | 81.19 | 0.81 | 0.84 | 7 | 7 | 6 | 7 | 27 |
| Model 2 | 0.90 | 87.90 | 0.88 | 0.91 | 5 | 5 | 6 | 7 | 23 | 1.39 | 69.74 | 0.88 | 0.73 | 1 | 2 | 7 | 4 | 14 |
| Model 3 | 1.01 | 84.80 | 0.85 | 0.87 | 2 | 2 | 4 | 3 | 11 | 1.15 | 79.37 | 0.79 | 0.80 | 5 | 5 | 4 | 6 | 20 |
| Model 4 | 0.70 | 90.53 | 0.91 | 0.92 | 8 | 8 | 8 | 8 | 32 | 0.84 | 89.68 | 0.90 | 0.96 | 8 | 8 | 8 | 8 | 32 |
| Model 5 | 1.01 | 84.73 | 0.85 | 0.88 | 2 | 1 | 4 | 4 | 11 | 1.14 | 79.50 | 0.80 | 0.84 | 6 | 6 | 5 | 7 | 22 |
| Model 6 | 0.96 | 86.40 | 0.86 | 0.90 | 4 | 4 | 5 | 6 | 19 | 1.38 | 69.43 | 0.71 | 0.76 | 2 | 1 | 2 | 5 | 10 |
| Model 7 | 0.85 | 89.20 | 0.89 | 0.88 | 7 | 7 | 7 | 4 | 25 | 1.32 | 71.82 | 0.73 | 0.84 | 3 | 3 | 3 | 7 | 16 |
| Model 8 | 0.86 | 89.05 | 0.89 | 0.91 | 6 | 6 | 7 | 7 | 26 | 1.19 | 78.56 | 0.79 | 0.80 | 4 | 4 | 4 | 6 | 18 |

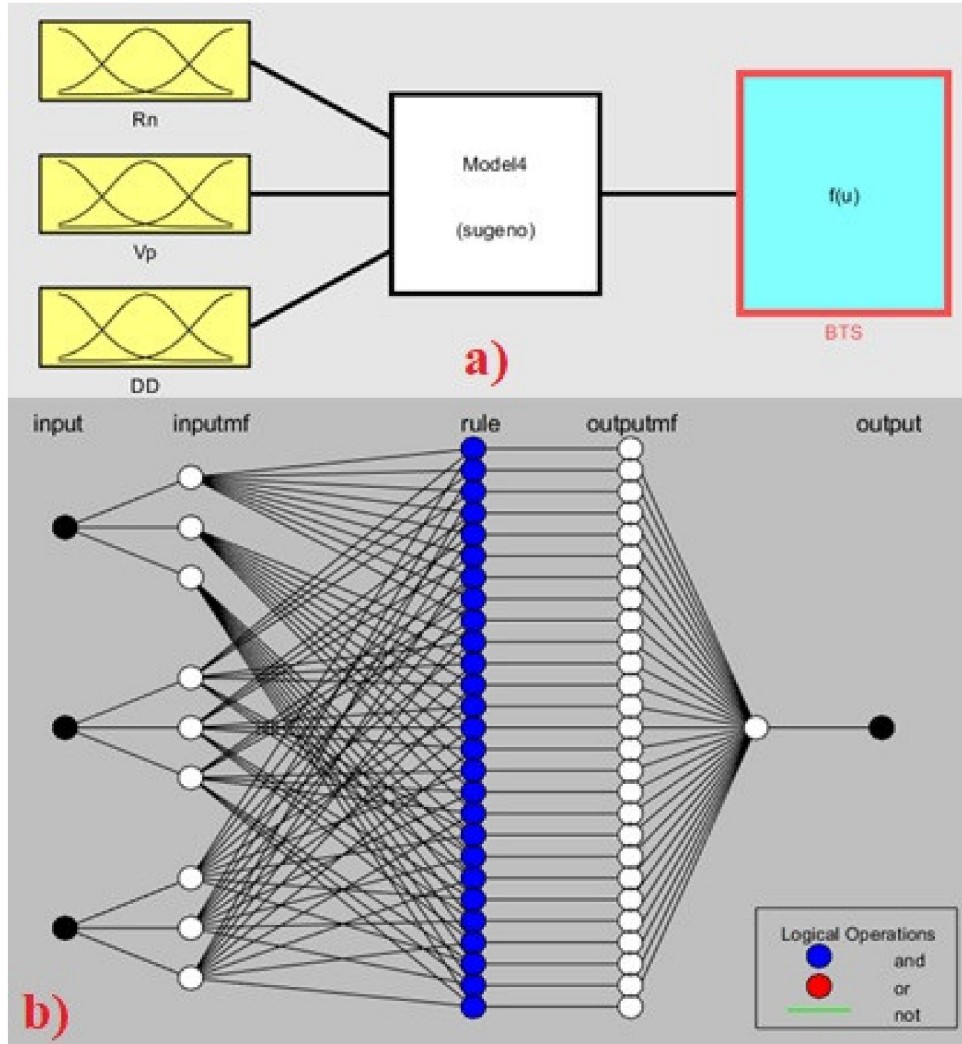

**Figure 5.** The proposed architecture used in this research: (**a**) base FIS model, (**b**) ANFIS structure.

**Table 10.** ANFIS Model 4 with its specifications.

| | |
|---|---|
| Number of Layers | 5 |
| Training Data Size | $102 \times 4$ |
| Testing Data Size | $25 \times 4$ |
| FIS Properties | Grid Partition |
| Input FIS Structure | |
|     MF Type | Gaussian |
|     MF Number | 333 |
| Output FIS Structure | |
|     MF Type | Constant |
| FIS Training | |
|     Optimum Method | Hybrid |
|     Error Tolerance | 0 |
|     Epochs Number | 88 |
| FIS System | |
|     Input | 3 |
|     Output | 1 |
|     Rules Number | 27 |

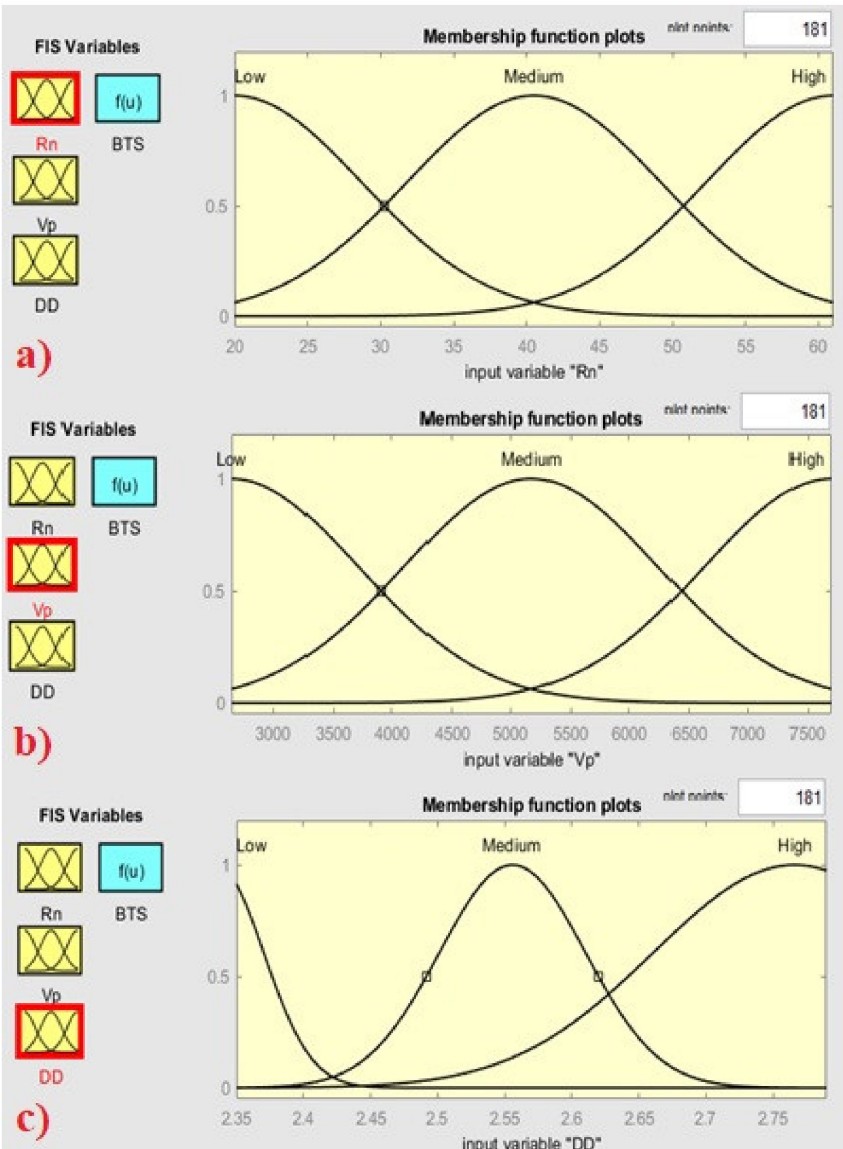

**Figure 6.** MFs plot of the selected ANFIS model: (**a**) $R_n$, (**b**) $V_p$, (**c**) DD.

## 4. Discussion

This section provides a quantitative assessment in terms of the performance of all established models during the testing and training phase. From the SR section, although the coefficient of determination results is within the acceptable range (based on previous studies), it did not yield a meaningful relationship with a strong level of accuracy. To improve prediction performance for BTS estimation, MLR and ANFIS modelling techniques were performed. As a result, four MLRs with multiple input parameters were proposed to enhance the BTS prediction significantly. Model MLR4 was able to generate a $R^2$ of 0.84 and 0.78 for the training and testing model, respectively. From this assessment, MLR $R^2$ (performance capacity) seems to be more reliable compared to the SR analysis with $R^2$ of 0.70 at most. It is important to mention that receiving a higher level of accuracy for predictive models is always of importance and interest in civil and mining engineering. Measured vs. predicted results of BTS for MLR4 are presented in Figure 7 for train and test stages. As shown, the prediction capacity was remarkably increased by the MLR model compared to SR models. However, higher $R^2$ or performance capacity values do not always imply that a model is superior. To evaluate the performance of the MLR model, the same dataset arrangement was employed using the ANFIS algorithm.

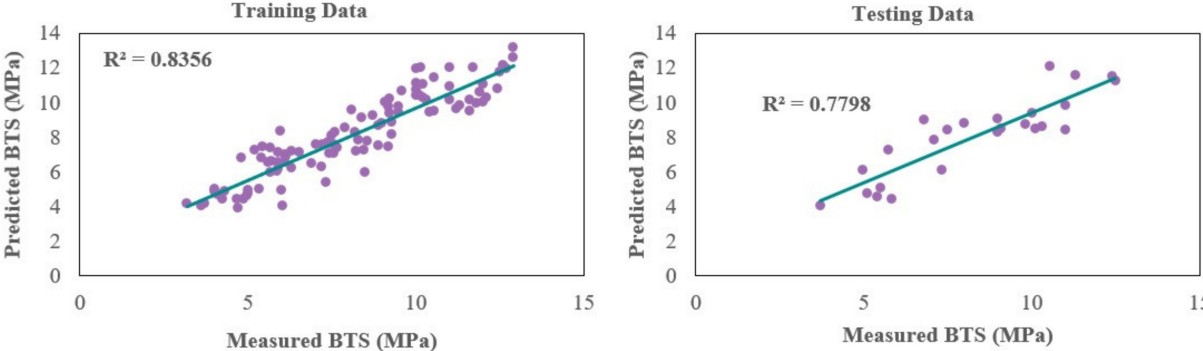

**Figure 7.** Measured vs. predicted BTS values by MLR Model 4.

The best predictive model will have the ideal best line of fit that minimizes the number of squares of divergence from the line of various data points [23]. Therefore, an outstanding predictive model will have the combination of lowest *RMSE* value and highest *VAF*, $R^2$, and a20-index values. Although the ANFIS model can run by using one training dataset, Al-Hmouz et al. [60] mentioned that the efficiency of the model can be improved when the testing datasets are combined with the training dataset to improve the accuracy of the model. The results of the ANFIS models showed that Model 4 was able to receive a high level of accuracy to predict tensile strength of rock material. The rule viewer of the proposed model displays a better visualization of the FIS structure (Figure 8). Based on the rule viewer, when the input parameter of $R_n$, $V_p$, and DD is 40.5, $5.17 \times 10^3$ m/s, and 2.57 g/cm$^3$, respectively, an output of BTS at 7.01 MPa is obtained. This figure also suggests that the ANFIS Model 4 has produced a total of 27 rules in which each rule has a single output MF, which is by default linear. The gradient vector was used to process the change in MF parameters, which shows how well the ANFIS is modelled by a given set of training data for a specific condition. FIS with many rules may generate a case-based reasoning model, in which each pair of training data have their own rules.

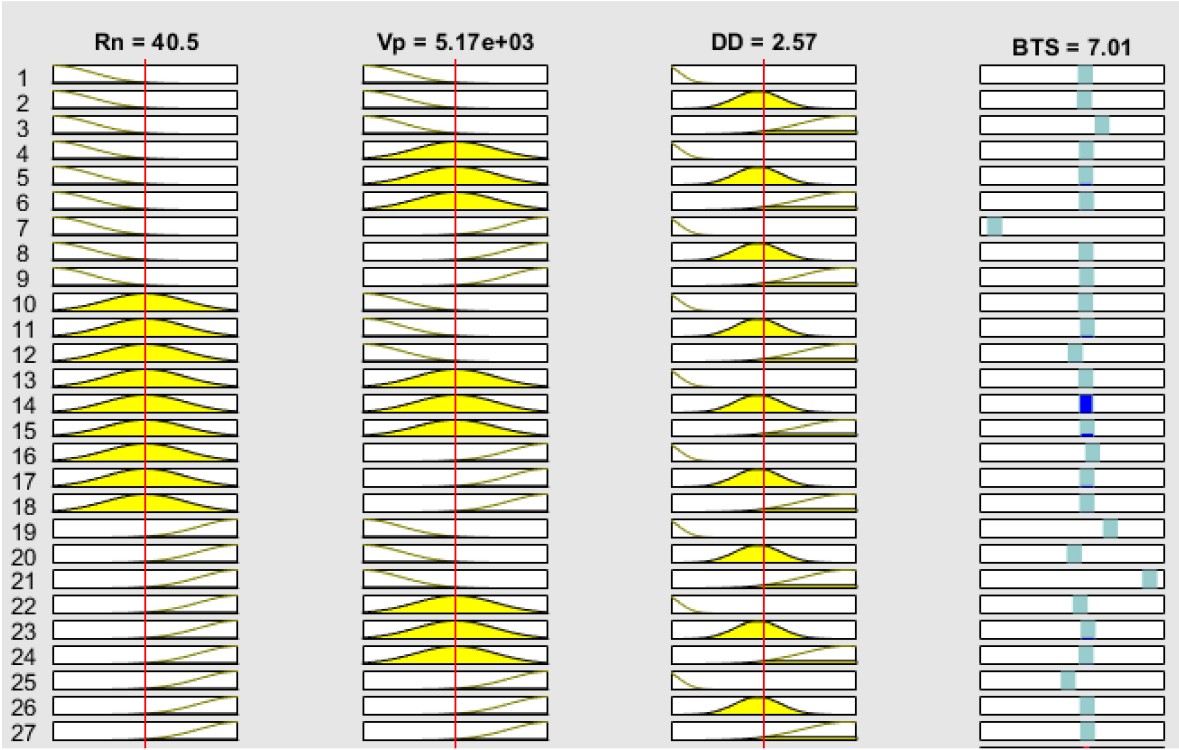

**Figure 8.** The integrated rules in the proposed ANFIS model.

The surface view tools generate and plot the output surface maps for the model which can be used to display the dependency of two inputs on the system output (Figure 9). For instance, high magnitude $R_n$ and $V_p$ will generate high BTS values (Figure 9a). Figure 9b shows that the model will generate a BTS approximation of 20 MPa when $R_n$ and DD of 60 and 2.7 g/cm$^3$ are being presented. Meanwhile, the plot clustering in Figure 9c indicates that the model produces higher BTS output with the allocation of higher $V_p$ and DD input values. Table 11 shows the results obtained by the selected MLR and ANFIS models to estimate BTS values. In addition, the graph of measured and predicted BTS values obtained by the best ANFIS model is displayed in Figure 10. According to Table 11 and Figure 10, it is obvious that the ANFIS model is able to increase prediction capacity of the MLR model in terms of all statistical indices. The clearer change is related to system error, which decreased from 1.05 to 0.69 in training and from 1.2 to 0.74 in testing. In addition, $R^2$ of the model improved from 0.84 to 0.93 and from 0.78 to 0.92 for training and testing stages, respectively. It is important to mention that full data used in the stage of testing are presented in Appendix A for better understanding.

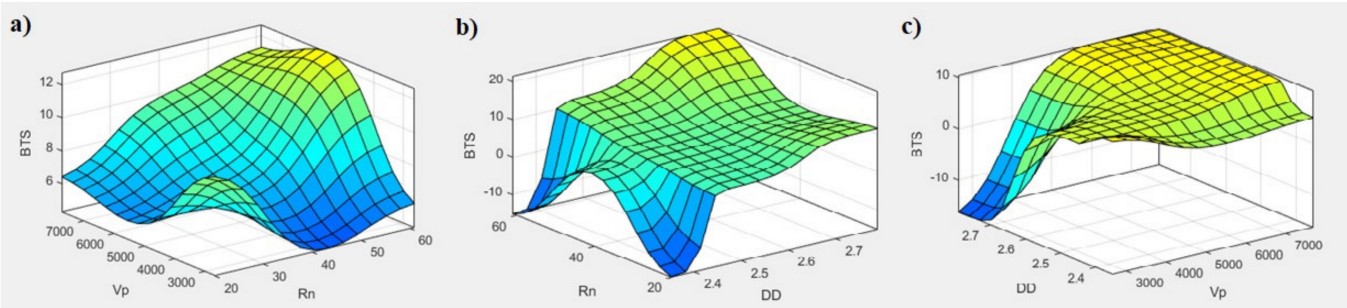

**Figure 9.** Surface visualization obtained from the selected ANFIS model. (**a**) BTS as a function of $R_n$ and $V_p$, (**b**) BTS as a function of $R_n$ and DD, and (**c**) BTS as a function of DD and $V_p$.

**Table 11.** The results obtained by the selected MLR and ANFIS models to estimate BTS values.

| Model Name | Train | | | | Test | | | |
|---|---|---|---|---|---|---|---|---|
| | *RMSE* | *VAF* (%) | $R^2$ | a20-Index | *RMSE* | *VAF* (%) | $R^2$ | a20-Index |
| ANFIS Model | 0.70 | 90.53 | 0.91 | 0.96 | 0.84 | 89.68 | 0.90 | 0.96 |
| MLR4 | 1.05 | 83.56 | 0.84 | 0.84 | 1.20 | 77.87 | 0.78 | 0.80 |

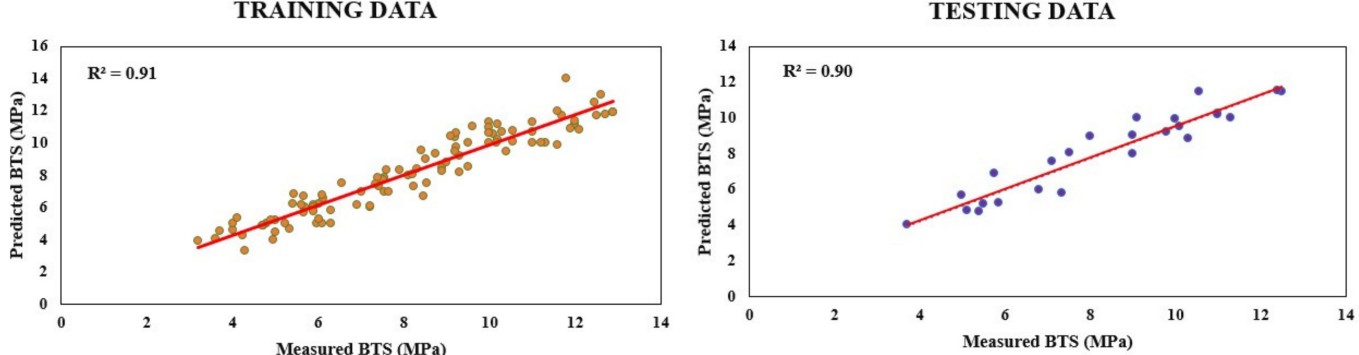

**Figure 10.** Measured vs. predicted BTS values by the proposed ANFIS model in this study.

A critical analysis of predictive modelling results by several researchers using soft computing techniques such as ANN was conducted to validate the outcomes of this research. For instance, the range of $R^2$ for an ANFIS predictive model proposed by Hasanipanah et al. [59] that integrates $R_n$, DD, and point-load index is between 0.857 to 0.897 for the testing stage of the model, which is lower than the performance prediction obtained in this

study. Ceryan et al. [43], who developed LS-SVM in predicting tensile strength of rock, obtained a $R^2$ of 0.86 which is lower than this study. In two other studies, Mahdiyar et al. [24] and Huang et al. [45] obtained similar results for their proposed models PSO-ANN and IWO-ANN, respectively. It is important to stress that most of the conducted studies in this field were focused on both non-destructive and destructive rock index tests. However, this study aimed to consider and use only non-destructive test results where the samples did not fail during or after these tests. By using these tests and the proposed structure of the ANFIS model in this study, similar results can be obtained by the other researchers and engineers.

## 5. Sensitivity Analysis

Sensitivity analysis (SA) which explores the relationship between a model's expectations and its model inputs, is useful for a computer-based framework. The multivariate nature of model inputs, as well as their uncertainty ranges, have a significant impact on systems. Conductive SA methodology can be beneficial for making more reliable prediction and allowing other researchers to make improvements in the future. Additionally, SA is able to identify the essential variable(s) that can give major influence on the predictive models. Hence, SA was carried out to recognize the relationship of each parameter with the ANFIS model. To apply this method, the following equation can be utilized to determine the relation strength ($r_{ij}$) between the model inputs ($R_n$, $V_p$, and DD) and output (BTS).

$$r_{ij} = \frac{\sum_{k=1}^{n} x_{ik} x_{jk}}{\sqrt{\sum_{k=1}^{n} x_{ik}^2 \sum_{k=1}^{n} x_{jk}^2}} \tag{13}$$

where $x_{ik}$ is the model input, $x_{jk}$ denotes the model output, and $r_{ij}$ indicates the strength of relation. Figure 11 shows the $r_{ij}$ values between each input and output parameter. The SA results demonstrate that $R_n$ is the most important factor for BTS prediction, followed by $V_p$ and DD. Similar results can be found in the SR and ANFIS techniques of the same study.

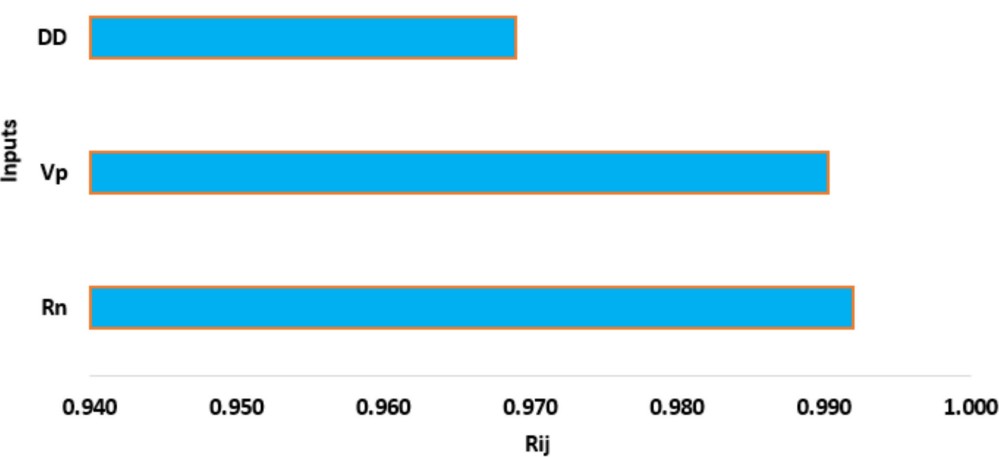

**Figure 11.** The importance of each input variable.

## 6. Limitations and Future Works

This research is subject to several limitations. Firstly, the database collected is predominantly made up of granite-type rock. According to Aydin and Basu [11], rock behaviours are site-specific, as they differ from one location to another. In this regard, the application of the recommended models to other types of rock that were not mentioned (such as basalts, marble, pumice, and so on) should be made with caution, as they might not yield the same

results as this study. It should also be noted that the model is appropriate to predict BTS when values of rock indices such as $R_n$ and $V_p$ are available in the same range as this research. It is suggested that future models should acquire data with larger sample sizes and variations to improve prediction accuracy. Among the various prediction techniques, this study focused on the capabilities of two conventional linear regression models (SR and MLR) and one form of artificial intelligence method (ANFIS). The implementation of optimization techniques such as IWO together with the ANFIS model can be considered in the future to examine their capability in predicting BTS values. In addition, a larger database comprising non-destructive rock index tests can be provided to propose a more comprehensive intelligence technique, since generalization of the proposed models is an important advantage in predictive models.

## 7. Conclusions

A comprehensive series of laboratory tests (i.e., non-destructive tests and Brazilian tests) were performed on more than 154 block samples brought from a water transfer tunnel project. Then, several SR-, MLR-, and ANFIS-based predictive models were designed and developed to assess the applicability of these methods in forecasting BTS. According to initial research, the need to develop BTS predictive models with higher degrees of accuracy was discovered through SR analyses. A range of 0.6–0.7 was recorded for the coefficient of determination of SR equations in predicting BTS values. Then, the authors decided to use another statistical-based technique which is able to consider the effects of all non-destructive tests as inputs in the analysis. From the regression statistic findings, the MLR4 model exhibited the best results with the highest-ranking scores of 28 among all other MLR models. The computed training RMSE, VAF, $R^2$ and a20-index values for this model were 1.05, 83.56, 0.84, and 0.84, respectively. The ANFIS model, on the other hand, significantly outperformed MLR analysis in terms of overall quality of the model. Model 4 of the ANFIS analysis achieved good model fit titles in which it ideally approximates the observed output. For this reason, the prediction using ANFIS Model 4 shall be introduced as the most robust approach in predicting BTS of granitic rock. In fact, the ANFIS structure proposed in this study, enjoying advantages of both ANN and fuzzy theory, can handle the BTS problem, which is complex and nonlinear. Proposing the ANFIS model, the accuracy of the MLR technique can be improved until 0.91 and 0.90 $R^2$ results are achieved. Additionally, based on sensitive analysis assessment, $R_n$ indicates the most effective input parameters on BTS of the rock. Meanwhile, DD provides the lowest impact on BTS with $R_{ij}$ value of 0.969.

**Author Contributions:** Conceptualization, D.J.A., A.A. and A.D.; methodology, F.N.S.H., D.J.A., Y.L.; software, F.N.S.H., D.J.A.; formal analysis, Y.L., F.N.S.H., D.J.A.; writing—original draft preparation, F.N.S.H., A.S.M., D.J.A.; writing—review and editing, Y.L., F.N.S.H., A.S.M., D.J.A., D.V.U., A.A., A.D.; supervision, D.J.A., D.V.U. All authors have read and agreed to the published version of the manuscript.

**Funding:** The research was funded by Act 211 Government of the Russian Federation, contract No. 02.A03.21.0011.

**Institutional Review Board Statement:** Not applicable.

**Informed Consent Statement:** Not applicable.

**Data Availability Statement:** The data are available upon request.

**Acknowledgments:** Authors of this study wish to express their appreciation to the University of Malaya for supporting this study and making it possible.

**Conflicts of Interest:** The authors declare no conflict of interest.

## Appendix A

**Table A1.** The used data in testing phase.

| Sample No. | $R_n$ | $V_p$ (m/s) | DD (g/cm$^3$) | BTS (MPa) |
|---|---|---|---|---|
| 1 | 33 | 4670 | 2.61 | 5.75 |
| 2 | 34 | 3210 | 2.53 | 7.34 |
| 3 | 52 | 6102 | 2.59 | 11 |
| 4 | 54 | 7155 | 2.68 | 12.5 |
| 5 | 43 | 5217 | 2.59 | 11 |
| 6 | 42 | 6635 | 2.6 | 6.8 |
| 7 | 33 | 6080 | 2.68 | 9.1 |
| 8 | 57 | 6980 | 2.68 | 12.4 |
| 9 | 40 | 7003 | 2.54 | 7.5 |
| 10 | 27 | 3615 | 2.42 | 5.4 |
| 11 | 22 | 3430 | 2.48 | 5.85 |
| 12 | 37 | 6005 | 2.65 | 10.3 |
| 13 | 40 | 5832 | 2.63 | 9.8 |
| 14 | 50.8 | 7152 | 2.76 | 11.3 |
| 15 | 28 | 3866 | 2.42 | 5.1 |
| 16 | 48 | 6503 | 2.57 | 10 |
| 17 | 38 | 6430 | 2.6 | 10.1 |
| 18 | 37 | 3050 | 2.38 | 5.5 |
| 19 | 43 | 6320 | 2.58 | 7.99 |
| 20 | 57 | 6659 | 2.76 | 10.55 |
| 21 | 28 | 5040 | 2.52 | 4.99 |
| 22 | 29 | 2870 | 2.37 | 3.7 |
| 23 | 40 | 5463 | 2.55 | 7.1 |
| 24 | 42 | 7002 | 2.59 | 8.99 |
| 25 | 33 | 5125 | 2.7 | 9 |

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
