# Peer review of "The Effects of Rock Index Tests on Prediction of Tensile Strength of Granitic Samples: A Neuro-Fuzzy Intelligent System"

_sustainability, doi:10.3390/su131910541_

Round 1

Reviewer 1 Report

In this paper, the effects of non-destructive rock index tests were examined through the use of multiple linear regression (MLR) and adaptive neuro-fuzzy inference system (ANFIS) approaches. Some parts are unclear. Discussion needs to be improved. 

The followings are the comments in detail:

1) What does “or rock material” mean in the tile?

2) The repeated content can be seen in the first paragraph of “1. Introduction” and “2. Significance of Study”.

3) In introduction, the motivation and important (difference) of this work should be further presented.

4) The part of “3.1 ANFIS Background” can be placed after “Laboratory Tests”.

5) The part of “3.2 Laboratory Tests” is too long.

6) What’s the full name of ISRM?

7) It’s better to put the test data as the appendix.

8) In the end of page 9, “Error! Reference source not found.” Please check.

9) From Fig. 10, R2 of predicted BTS values is just little greater than that of measured values according to ANFIS model.

Author Response

Response to Reviewer #1

The authors would like to thank the reviewer for his/her time spent to review our manuscript and to provide the positive feedbacks. The quality of this paper has surely been improved by addressing the comments summarized as follows:

Comment 1: In this paper, the effects of non-destructive rock index tests were examined through the use of multiple linear regression (MLR) and adaptive neuro-fuzzy inference system (ANFIS) approaches. Some parts are unclear. Discussion needs to be improved.

Reply: Thank you for your time to review our paper. We have revised our paper based on your comments. Please consider our changes in revised manuscript.

The followings are the comments in detail:

Comment 2: What does “or rock material” mean in the title?

Reply: Thank you for this comment.  As you know, we have 2 terms related to rock as geomaterial which are rock material and rock mass. Normally, the samples collected from field or from mass of rock in order to investigate properties of the rock, called as rock material. The same that the authors of this study did. Because of that, we used the term of rock material in the title.

Regarding or, it should be “of”. We apologize for this mistake.

We changed the title as follows for better understanding and more general:

“The effects of rock index tests on prediction of tensile strength of granitic samples: a neuro-fuzzy intelligent system”

Comment 3: The repeated content can be seen in the first paragraph of “1. Introduction” and “2. Significance of Study”.

Reply: Thank you for mentioning this valuable comment. Based on this comment, we have removed section 2 (Significance of Study) from the revised manuscript. Please check it in our revised manuscript.

Comment 4: In introduction, the motivation and important (difference) of this work should be further presented.

Reply:  Thank you for mentioning this comment. We have mentioned the mentioned points at the end of introduction section. Please check it in our revised manuscript.  

Comment 5: The part of “3.1 ANFIS Background” can be placed after “Laboratory Tests”.

Reply: We appreciate your comment. We have revised this section based on this comment. Please check this section in our revised manuscript.

Comment 6: The part of “3.2 Laboratory Tests” is too long.

Reply: Thank you for this comment. We have removed some parts and totally more than 200 words were removed from this section. Please check it in our revised manuscript.

Comment 7: What’s the full name of ISRM?

Reply: Thank you. It is actually, International Society for Rock Mechanics, it was added in the revised manuscript.   

Comment 8: It’s better to put the test data as the appendix.

Reply: Thank you and done.

Comment 9: In the end of page 9, “Error! Reference source not found.” Please check.

Reply: We appreciate this point. We have checked all references and corrected accordingly.  

Comment 10: From Fig. 10, R2 of predicted BTS values is just little greater than that of measured values according to ANFIS model.

Reply: We appreciate your comment. Please consider our explanations as follows:

In this study, we successfully showed an improvement in prediction results of BTS from simple regression step and then multiple regression technique and finally, ANFIS model. Figure 10 shows only results of ANFIS model for training and testing stages. As can be seen from this figure, R2 of ANFIS model is as 0.91 and 0.90 was obtained for train and test stages, respectively. These values were obtained as 0.84 and 0.78 respectively, for multiple regression model. It is confirmed that the ANFIS model is able to increase prediction capacity significantly and is the best model proposed in this study.

Thank you for your time and kind consideration.

Best regards,

Danial Jahad Armaghani

Reviewer 2 Report

It is seen as a research that predicts using artificial intelligence techniques using the recent trend of big data. I think it is an interesting topic to predict tensile strength from Schmidthammer test results using neurofuzzy intelligent systems. The process of data collection, processing, and result derivation is well-structured and well described. As the author described in relation to the concept of the paper, tensile strength is one of the important mechanical properties. In general, the strength value is obtained through an indoor experiment with a sample representative of the target rock, but it is questionable whether predicting from the Schmitthammer strength value attempted in this paper will yield more reliable results. I would like to hear the author's opinion on this part. If it has already been mentioned in the paper, please let me know.

Author Response

Response to Reviewer #2

The authors would like to thank the reviewer for his/her time spent to review our manuscript and to provide the positive feedbacks.

Comment 1: It is seen as a research that predicts using artificial intelligence techniques using the recent trend of big data. I think it is an interesting topic to predict tensile strength from Schmidt hammer test results using neurofuzzy intelligent systems. The process of data collection, processing, and result derivation is well-structured and well described. As the author described in relation to the concept of the paper, tensile strength is one of the important mechanical properties. In general, the strength value is obtained through an indoor experiment with a sample representative of the target rock, but it is questionable whether predicting from the Schmidt hammer strength value attempted in this paper will yield more reliable results. I would like to hear the author's opinion on this part. If it has already been mentioned in the paper, please let me know.

Reply: Firstly, thank you very much for your time to review our paper and also for your positive feedback. Regarding your comment, we would like to express that the objective of this study is to use non-destructive tests from all available rock index tests for predicting tensile strength of the rock. As you know, one of the famous non-destructive tests in rock mechanics is Schmidt hammer test. You are right, sometimes; this test is not a good parameter for predicting strength parameters of rock samples especially for predicting uniaxial compressive strength. However, we have several published works reporting a good correlation between this parameter and rock strength. In this study, we received a very good result for correlation between Rn and BTS in a form of a linear equation as presented in Figure 4. The result of R2 = 0.704 confirmed that we can get a reliable values for BTS if we use only one input/prediction. The rest of modeling in this study was in line to propose models where the highest performance prediction is of interest. Therefore, we moved to multiple regression and neuro-fuzzy models. The simple regression part can be used if an average level of model accuracy and prediction is needed.

Thank you for your time and kind consideration.

Best regards,

Danial Jahad Armaghani

Reviewer 3 Report

This study presented a series of laboratory tests and applied several mathematical models (SR, MLR, and ANFIS) to assess the applicability of these methods in forecasting BTS. This is an important research area and the results show some interesting contributions to this field. This reviewer would like to forward the following comments:

  1. The rationales of SR, MLR, and ANFIS methods should be given.
  2. Why not compressive strength because it is in general employed as an index to evaluate brittle materials (like rocks)?
  3. Rocks’ compressive strengths are much greater than their tensile strengths. Hence, from a practical viewpoint, it is more desirable to utilize rocks’ under compressive loading instead of tensile forces. Please comment on it.

Upon addressing the aforementioned comments, this manuscript may be considered for publication. Thus, a major revision is recommended.

Author Response

Response to Reviewer #3

The authors would like to thank the reviewer for his/her time spent to review our manuscript and to provide the positive feedbacks.

Comment 1: This study presented a series of laboratory tests and applied several mathematical models (SR, MLR, and ANFIS) to assess the applicability of these methods in forecasting BTS. This is an important research area and the results show some interesting contributions to this field. This reviewer would like to forward the following comments:

The rationales of SR, MLR, and ANFIS methods should be given.

Reply: Firstly, thank you very much for your time to review our paper and also for your  positive feedback. Regarding the mentioned comment, it is important to note that we followed a typical flow in simulation and prediction studies. They normally start with a SR technique which is easy to conduct, but it is at the same time not accurate enough to solve the problem. Then, we moved to a multiple regression model which more than one predictor can be used to increase prediction capacity. Finally, an intelligence technique based on a combination of ANN and fuzzy models, was used to predict BTS values. We added similar explanations in the revised manuscript and section 2.3. Please check it.

Comment 2: Why not compressive strength because it is in general employed as an index to evaluate brittle materials (like rocks)?

Reply: Thank you for this valuable comment. You are right, in many projects, compressive strength of the rock is of importance and essential. There are also many published studies in this regard and to predict compressive strength of rock material [1–4]. However, tensile strength of rock material is important in some projects related to rock excavation like tunneling and mechanized tunneling. In addition, there are a few studies only published in the area of BTS prediction. Therefore, the authors of this study decided to propose a prediction model for estimating BTS values.

  1. Moradian ZA, Behnia M (2009) Predicting the uniaxial compressive strength and static Young’s modulus of intact sedimentary rocks using the ultrasonic test. Int J Geomech 9:14–19
  2. Grima MA, Babuška R (1999) Fuzzy model for the prediction of unconfined compressive strength of rock samples. Int J Rock Mech Min Sci 36:339–349
  3. Ghasemi E, Kalhori H, Bagherpour R, Yagiz S (2018) Model tree approach for predicting uniaxial compressive strength and Young’s modulus of carbonate rocks. Bull Eng Geol Environ 77:331–343
  4. Mohamad ET, Armaghani DJ, Momeni E, Abad SVANK (2015) Prediction of the unconfined compressive strength of soft rocks: a PSO-based ANN approach. Bull Eng Geol Environ 74:745–757

Comment 3: Rocks’ compressive strengths are much greater than their tensile strengths. Hence, from a practical viewpoint, it is more desirable to utilize rocks’ under compressive loading instead of tensile forces. Please comment on it.

Reply: We appreciate your kind comment. As I mentioned previously, you are completely right. Results of compressive strength are much greater than those results of tensile strength. As presented in Table 3, the average values of 8 MPa was obtained for tensile strength values, while an average range of 90-100 MPa was obtained for the same samples under compressive pressure. This significance difference is related to behavior of rock material which is considered as brittle one. The same behavior can be seen in concrete material. In some projects like rock excavation, this brittleness index of the rock will be highlighted [5]. Therefore, more studies are needed to investigate the behavior of rock strength under tensile strength. This will help to have a more accurate model to predict the performance of excavation rate in tunneling projects or in surface excavation projects.

  1. Yagiz S (2009) Assessment of brittleness using rock strength and density with punch penetration test. Tunn Undergr Sp Technol 24:66–74

Upon addressing the aforementioned comments, this manuscript may be considered for publication. Thus, a major revision is recommended.

Reply: Thank you again for reviewing our paper and valuable comments. We hope our responses are strong enough to satisfy you.  

Thank you for your time and kind consideration.

Best regards,

Danial Jahad Armaghani

Round 2

Reviewer 1 Report

The manuscript has been revised.

Reviewer 3 Report

Accept for publication. Thank you for the reply.